# Electron cryo-microscopy of bacteriophage PR772 reveals the elusive vertex complex and the capsid architecture

Hemanth KN Reddy[1]*, Marta Carroni[2], Janos Hajdu[1,3], Martin Svenda[1,4]*

[1]Department of Cell and Molecular Biology, Laboratory of Molecular Biophysics, Uppsala University, Uppsala, Sweden; [2]Department of Biochemistry and Biophysics, Stockholm University, Stockholm, Sweden; [3]Institute of Physics, ELI Beamlines, Academy of Sciences of the Czech Republic, Prague, Czech Republic; [4]Department of Applied Physics, Biomedical and X-Ray Physics, KTH Royal Institute of Technology, Stockholm, Sweden

**Abstract** Bacteriophage PR772, a member of the *Tectiviridae* family, has a 70 nm diameter icosahedral protein capsid that encapsulates a lipid membrane, dsDNA, and various internal proteins. An icosahedrally averaged CryoEM reconstruction of the wild-type virion and a localized reconstruction of the vertex region reveal the composition and the structure of the vertex complex along with new protein conformations that play a vital role in maintaining the capsid architecture of the virion. The overall resolution of the virion is 2.75 Å, while the resolution of the protein capsid is 2.3 Å. The conventional penta-symmetron formed by the capsomeres is replaced by a large vertex complex in the pseudo T = 25 capsid. All the vertices contain the host-recognition protein, P5; two of these vertices show the presence of the receptor-binding protein, P2. The 3D structure of the vertex complex shows interactions with the viral membrane, indicating a possible mechanism for viral infection.

DOI: https://doi.org/10.7554/eLife.48496.001

**\*For correspondence:**
hemanth.kumar@icm.uu.se (HKNR);
Martin.Svenda@icm.uu.se (MS)

**Competing interests:** The authors declare that no competing interests exist.

## Introduction

Bacteriophage PR772 is a double-stranded DNA (dsDNA) virus with a 70 nm diameter icosahedral protein capsid encapsulating the internal lipid bilayer along with numerous proteins and a 15 kbp long linear genome. It belongs to the *Tectiviridae* family and infects gram negative hosts like *Escherichia coli*, *Salmonella typhimurium* and other bacteria, carrying a R772 plasmid-encoded receptor complex through which DNA can be transported during bacterial conjugation (*Coetzee et al., 1979*; *Coetzee and Bekker, 1979*; *Mäntynen et al., 2019*; *Reddy et al., 2017*). Much of the functional knowledge of the viral proteins is inferred from previous studies on PRD1, a close relative of PR772 with a genome sequence identity of 97.2% (*Lute et al., 2004*; *Saren et al., 2005*). The most striking features of phages from those members of the *Tectiviridae* family that infect Gram-negative bacteria, is the presence of an inner lipid membrane and lack of a tail in the dormant viral particle. During the process of infection, these viruses produce a membranous tube derived from the inner membrane of the viral particle, lined from the inside by proteins P7, P11 and P32. This tube is used to inject the viral dsDNA into the host (*Peralta et al., 2013*; *Saren et al., 2005*).

Genome analysis of PR772 identified 32 open reading frames (ORFs) containing at least 40 codons (*Lute et al., 2004*). Twenty-eight annotated proteins are known to be expressed from the genome, of which three do not make it into the final assembly (*Butcher et al., 2012*; *Lute et al.,*

*2004*). In previous studies on PRD1, it was shown that the capsid is formed by proteins P3 (major capsid protein), P30 (capsid associated protein) and the vertex complex. The vertex complex includes the penton protein (P31), the host-recognition protein (P5) and the receptor-binding protein (P2) along with the infectivity protein (P16) that acts as a cementing protein, holding the vertex complex together and also stabilizing the pseudo-icosahedral capsid of the virus (*Saren et al., 2005*). Proteins P6, P9, P20 and P22 are involved in DNA packaging and proteins P7, P14, P11, P18, P32 and P34 are responsible for DNA delivery to the host (*Grahn et al., 2002*; *Lute et al., 2004*). The *Tectiviridae* family of viruses are also known to have structural similarities to adenovirus (*Saren et al., 2005*).

In previous studies on bacteriophage PRD1, many models have been proposed to explain the architecture of the penton base and the vertex complex. Most of these models were speculations based on the observations from gene mutation/knock-out and in-vitro studies of proteins P31, P5 and P2 from PRD1. The in-vitro protein expression studies showed that P31 forms pentamers and P5 form multimers of trimers (i.e, $(P5_3)_1$, $(P5_3)_2$, $(P5_3)_3$) (*Sokolova et al., 2001*). The gene knock-out studies showed that P31$^-$ mutants produced incomplete PRD1 particles that lacked P31, P5 and P2 functions. They failed to form the vertex complex. A P5$^-$ mutant produced intact viral particle but lacked both P5 and P2 functions (*Rydman et al., 1999*). These observations led to the hypothesis that the vertex complex appears to be a single spike formed by a pentameric P31 base binds the trimeric P5 spike protein to which P2 is bound (*Huiskonen et al., 2007*; *Caldentey et al., 2000*; *Rydman et al., 1999*; *Sokolova et al., 2001*). Later, a combination of SAXS modeling of the P5 protein and a low resolution cryoem study of the vertex region, showed that P2 and P5 form two spikes, not one, as previously described. Based on the low-resolution SAXS model of P5, it was speculated that the N-terminal base of P5 could have a size similar to P31 due to sequence similarity (*Huiskonen et al., 2007*). With the available experimental data, the generally accepted composition of the vertex complex of *Tectiviridae* is that P31 is a homopentamer and forms the penton base. P5 and P2 are attached to the P31 penton base but their arrangement is not known. (*Butcher et al., 2012*; *Huiskonen et al., 2007*). The high resolution x-ray crystallographic structure of the P2 protein is known but the location and orientation with respect to the penton base in its functional form is currently not known (*Xu et al., 2003*). Putatively, it is suggested that the beta-propeller motif of P2 might be involved in binding to the host receptor.

Here we present the high-resolution structure of bacteriophage PR772 using electron cryo-microscopy (CryoEM). The high-resolution map helped us to resolve subtle variations in the protein conformations and their influence on the formation of a viable viral particle. The N-terminal region of the P3 subunits have three conformations, not two as previously described (*Abrescia et al., 2004*). The new N-terminal P3 conformation plays an important role in accommodating the P30 protein during particle assembly. The C-terminal region of P3 not only shows the formation of a β-sheet with P30 but also helps in locking the adjacent trisymmetrons through a hinge mechanism, thus facilitating the formation of icosahedral particles and regulating their size. Localized asymmetric reconstruction of the vertex region of PR772 revealed a P5-P31 heteropentameric base and the binding of P2 to P5 in the complex. A combination of high-resolution icosahedral symmetrized single-particle reconstruction, localized asymmetric reconstruction and focused classification has enabled us to answer some of the intriguing questions about the particle architecture, composition of the penton base and arrangement of the vertex complex.

## Results

### High resolution capsid map at 2.3 Å

The structure of bacteriophage PR772 was determined by electron cryo-microscopy. The overall resolution determined by Fourier shell correlation (FSC) @0.143 was 2.75 Å (*Figure 1—figure supplement 1*). The local resolution estimated using the two unfiltered final half maps with ResMap (*Kucukelbir et al., 2014*) showed that most of the capsid was resolved to 2.3 Å (*Figure 1A and B*). The resolution of the regions that interact with the inner lipid bilayer was lower at about 3.2 Å (*Figure 1C*). The areas around the icosahedral five-fold axes had resolutions varying between 2.3–3.0 Å (*Figure 1D–F*). At a root mean square deviation (RMSD; deviation away from noise as visualized in Coot, where noise is 0) of 4.2, the side chains of the amino acid residues were visible for

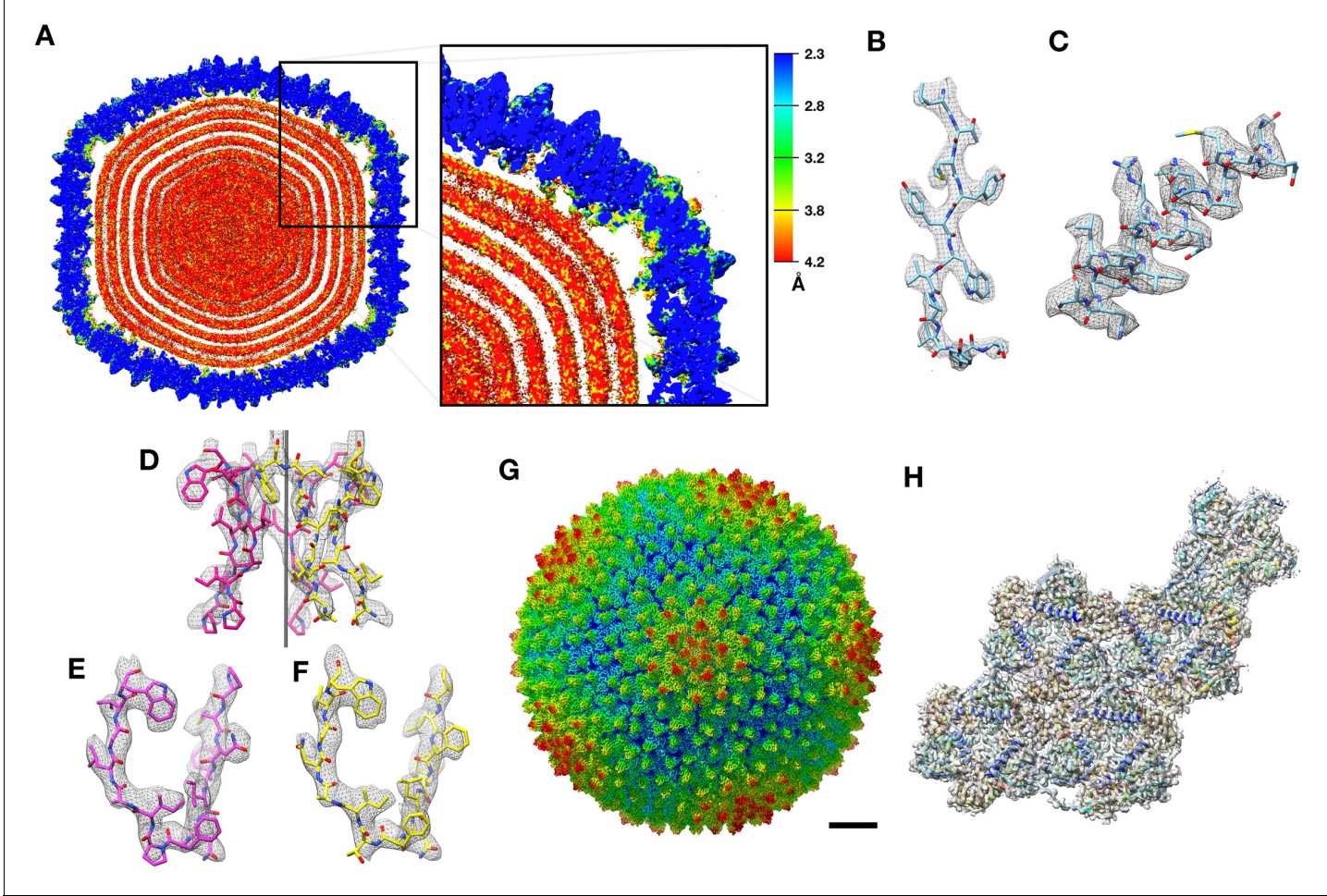

**Figure 1.** CryoEM structure of PR772 and resolution estimates. (A) The local resolution estimate of the CryoEM map from ResMap. The map shows the distribution of resolution in different regions. (Visualized using USCF Chimera, with volume viewer parameters: Style surface, step 1 and level 0.037, Plane 418, Axis Y, Depth 23). Most of the capsid that was used for model building is resolved at 2.3 Å. (B, C and D) Show the quality of the map in different regions. (B) Quality of the map at the core of the capsid protein P3 (Chain B, residues 162–173) where the local resolution estimate is 2.3 Å. (C) Quality of the map close to the membrane (P3 Chain B, residues 18–35) where the resolution is estimated to 3.2 Å. (D) Quality of the map close to the five-fold vertex of the icosahedral viral particle. The black vertical line represents the five-fold axis. (E and F) The initial model fit of P5 residues 108-121 (in pink) and P31 residues 113–126 (in yellow) to the same region of the map using Phenix: Find helix and sheets with respective protein sequences as input. (G) The post processed map of PR772 and the scale bar represents 10 nm. (H) The map:model fit of the asymmetric unit as seen from the inside of the viral particle.

DOI: https://doi.org/10.7554/eLife.48496.002

The following source data and figure supplement are available for figure 1:

**Source data 1.** Relion postprocess star file.
DOI: https://doi.org/10.7554/eLife.48496.005

**Figure supplement 1.** FSC curves of the overall resolution estimate of the 3D reconstruction (green: unmasked map, blue: masked map, black: FSC corrected, red: phase randomized map).
DOI: https://doi.org/10.7554/eLife.48496.004

most of the capsid region. The inner membrane layers of lipid, protein and dsDNA were smeared due to averaging and symmetry mismatch. The resolution in most of the regions was high enough (3.2 Å – 2.3 Å) to build a de-novo model of the asymmetric unit comprising of P3 (*Figure 2—figure supplement 1*), P30 (*Figure 2—figure supplement 2*), P5 (*Figure 4—figure supplement 3*), P31 (*Figure 4—figure supplement 3*) and P16 (*Figure 6—figure supplement 1*) into the icosahedrally averaged CryoEM map (*Figure 1H* and *Table 1*) (*Supplementary file 1*).

**Table 1.** Data collection, Processing and Model refinement parameters.

| Parameter | Value |
| --- | --- |
| **Data collection** | |
| Voltage (kV) | 300 |
| Magnification (x) | 130000 |
| Å/pix | 1.06 |
| Energy Filter with Slit (eV) | 20 |
| Frames per Micrograph | 40 |
| Total Dose ($e^-/Å^2$) | 40 |
| Micrographs | 3220 |
| Defocus Range (μm) | 0.8–2.6 |
| **Data Processing** | |
| Micrographs | 3200 |
| Frames used | 4–40 |
| Å/pix | 1.06 |
| Particles (Total) | 46348 (56275) |
| Symmetry Applied | I4 |
| Overall Resolution@$FSC_{0.143}$ (Å) | 2.75* |
| B-factor ($Å^2$) | −104.93* |
| **Model Refinement** | |
| Composition | |
| Chains | 19 |
| Atoms | 41731 (Hydrogens: 0) |
| Refinement | |
| $CC_{mask}$ | 0.7961 |
| $CC_{volume}$ | 0.7941 |
| ADP (B-factors) | |
| Iso/Aniso | 41731/0 |
| Mean | 101.95 |
| RMS deviations | |
| Bonds (Å) | 0.006 |
| Angles (°) | 0.806 |
| EMRinger Score | 5.40 |
| MolProbity validation | |
| Clash score, all atoms | 3.35 |
| MolProbity score | 1.57 |
| Rotamer Outliers (%) | 0.16 |
| Ramachandran | |
| Favoured (%) | 93.04 |
| Allowed (%) | 6.83 |
| Outliers (%) | 0.13 |

* Calculated by RELION

DOI: https://doi.org/10.7554/eLife.48496.003

The CryoEM 3D reconstruction of PR772 shows that the viral particle follows a pseudo T = 25 lattice architecture with a (h,k) of (0,5) resulting in an icosahedral structure with 20 large trisymmetrons and 12 penta-symmetrons (*Caspar and Klug, 1962*; *Wrigley, 1969*). However, the analysis revealed that the penta-symmetrons were hetero-pentamers. Each of these trisymmetrons have 36 copies of P3, the major capsid protein (MCP), arranged as 12 trimers, each of which structurally appears to be hexagonal in shape. At the fivefold vertices, typical penta-symmetrons are replaced by vertex complexes to complete the icosahedral shell. With a pseudo T = 25 architecture, PR772 is one of the larger wild type viruses resolved to a resolution below 3 Å.

## Major Capsid Protein and its conformations

P3 is the major capsid protein, which builds up the trisymmetrons of the capsid, and it is the most abundant protein found in bacteriophage PR772 (*Figure 2A*). The P3 monomers (subunits a, b and c) are interlocked to form a trimer that appears as a hexon (*Figure 2B–K*). They exhibit the double-barrel trimer arrangement, as previously seen in viruses of the adenovirus linage (*Benson et al., 1999*; *Reddy et al., 2010*). The Leu130 - Ala150 loops from subunits a, b and c of P3 interact with each other in a cyclic manner at the centre of the trimer complex to stabilize it (*Figure 2*). The monomers have a similar structure in bulk, with minor differences to accommodate the more significant variations in the C and N-terminal conformations. The asymmetric unit has four such unique trimers along with P30, P16 and a penton protein. *Figure 2B–K* show the four unique trimers and their subunit arrangement.

The N-terminal of the P3 monomers have three different conformations; a helix turn helix, a long helix and a long helix with a kink (*Figure 2—figure supplement 1*). *Figure 2D* is a good schematic to visualize all N-terminal conformations in a single trimer (subunit a shows the helix-turn helix, subunit b shows the long helix and subunit c shows the helix with a kink). When the N-terminal adopts a helix turn helix, the shorter helix close to the N-terminal is bent away from the lipid membrane and interacts with the adjacent subunit of the trimer. The long helix with a kink behaves similar to a helix turn helix where the kink twists the helix away from the membrane but it is not embedded deeply into the adjacent subunit of the trimer. To accommodate the helix turn helix or the helix with a kink in the P3 subunits, the loop formed by residue Tyr351 – Val358 in these subunits is flipped. In case of the long helical conformation, the N-terminal residues Met1 – Gln6 anchor the P3 subunit to the lipid membrane (*Figure 3A*).

The N-terminal region of subunit a is more flexible as compared to the same region of subunits b or c in the four unique trimers. The N-terminal of subunit a can form either a helix turn helix bent away from the membrane (*Figure 2D–E*) or a long helix that anchors the subunit to the membrane (*Figure 2C,F*). The N-terminal region of subunit c either forms a helix turn helix (*Figure 2C,E,F*) or a long helix with a kink (*Figure 2D* and *Figure 3B*). The N-terminal of the b subunits always forms a long helix that anchors the subunit to the membrane (*Figure 2C–F* and *Figure 3A*). Subunit b of trimer 1, which is present close to the penton has the Tyr351 – Val358 loop flipped even though there is no helix turn helix or a long helix with a kink in this region that must be accommodated. As it will be shown later, this loop accommodates P30 (*Figure 3B* and *Figure 2—figure supplement 3*).

The C-termini are more variable as compared to the N-termini of the trimers. They adopt four different conformations; one conformation is a long strand extending towards the lipid membrane and found in subunit b of all trimers (*Figure 2C–F,H–K*). The second conformation extends away from the membrane into the peripheral space between the trimers and it is also the most common C-terminal conformation. Two of the four instances in subunit a (*Figure 2D,E,I,J*) and two of the four instances in subunit c (*Figure 2E,F,J,K*) adopt this conformation. The third conformation is seen in subunit a of trimer 1, where the C-terminal runs parallel to the long N-terminal helix of the same subunit and it is embedded into the adjacent trimer 1 of the neighboring trisymmetron (*Figure 2C,H*). The fourth type of C-terminal conformation is seen in 2 instances of subunit c when they reside close to P30 (*Figure 2C,D,H,I*). Here, the C-terminal is elongated and runs towards the lipid membrane, grazing it.

## Penton base is a heteropentamer of P5 and P31

The penton region of the icosahedrally symmetrized CryoEM map of PR772 was sectioned from the whole viral map using UCSF Chimera (*Goddard et al., 2007*). To generate an initial model for de-

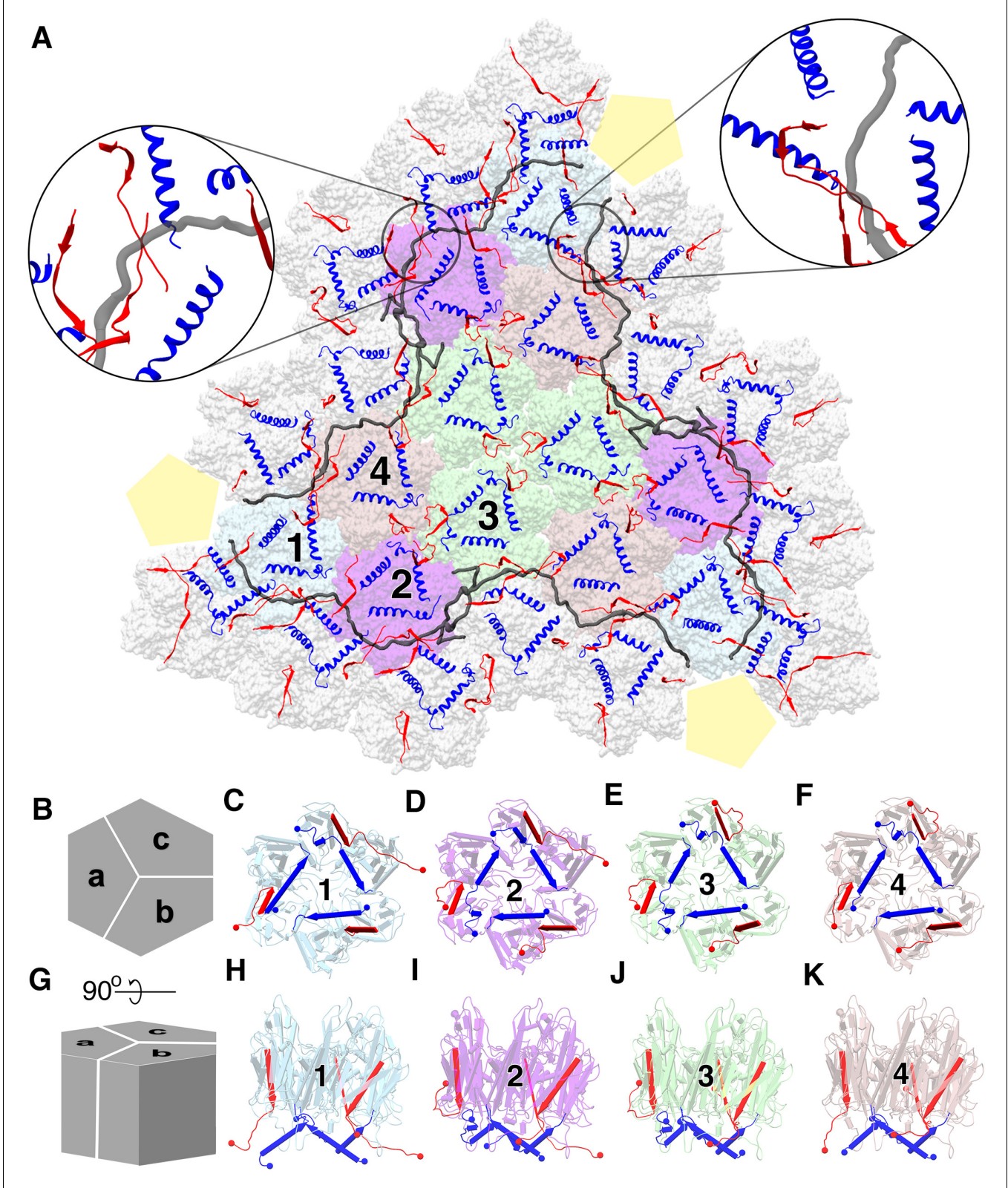

**Figure 2.** Major capsid protein P3 and its conformations. (**A**) (visualized from the outside of the viral particle) The four unique P3 trimers (represented in four different colors) and their arrangement forming the trisymmetron bound by the P30 dimers (in gray). The C-terminal region and the N-terminal region of P3 subunits are colored in red and blue respectively. The highlighted regions show the locking of P30 (in black) by the C-terminal region of the neighboring P3 subunits, leading to the formation of a hinge-like mechanism which is not seen in PRD1. (**B**) Schematic representation of the P3

*Figure 2 continued on next page*

*Figure 2 continued*

trimers and the subunit arrangement to form a hexagonal capsomer (as viewed from outside) and (C), (D), (E,) (F) are aligned to this view. (G) It is the orthogonal view to the schematic (B) and (H), (I), (J), (K) are aligned to this view. Different views of trimer 1 (C,H), trimer 2 (D,I), trimer 3 (E,J) and trimer 4 (F,K) show the variation in the N-terminal (shown as blue cylinders with the arrow heads pointing towards the C-terminal) and C-terminal (shown as red planks with the arrow heads pointing towards the C-terminal) region of P3 subunits. N and C termini are shown as spheres with respective colors. They are colored to match (A). The yellow pentagons are a schematic representation of the penton.

DOI: https://doi.org/10.7554/eLife.48496.006

The following figure supplements are available for figure 2:

**Figure supplement 1.** Shows the different N-terminal conformations of the P3 monomer.

DOI: https://doi.org/10.7554/eLife.48496.007

**Figure supplement 2.** Modelled P30 protein and its map density fit.

DOI: https://doi.org/10.7554/eLife.48496.008

**Figure supplement 3.** Variations in the P3 subunits.

DOI: https://doi.org/10.7554/eLife.48496.009

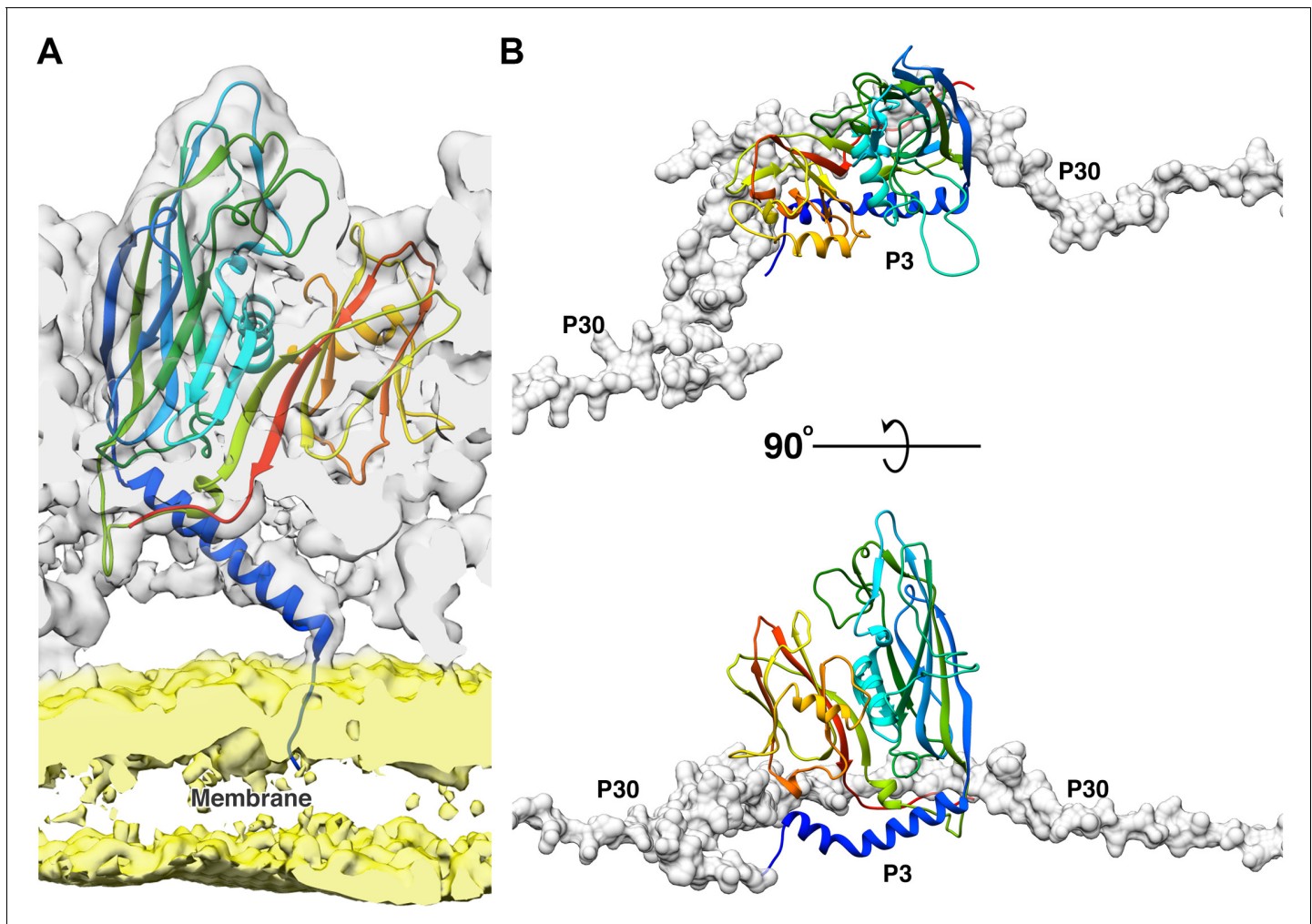

**Figure 3.** Function of 2 N-terminal helices of the P3 subunits. (A) Shows the model of the long N-terminal helix conformation (blue) of the P3 subunit anchoring to the membrane. The CryoEM map was low pass filtered to 5 Å to reduce noise in the membrane region due to map sharpening. In the CryoEM map, the capsid region is shown in gray and the membrane region is shown in yellow. (B) The newly discovered P3 N-terminal conformation, a long helix with a kink accommodates the P30 dimer (shown in gray, rendered as surface).

DOI: https://doi.org/10.7554/eLife.48496.010

novo model building of the penton region, PHENIX (*Adams et al., 2010*): Find Helices and Strands was used with both RESOLVE and PULCHAR options enabled. Due to the high sequence similarity between P31 and the N-terminal domain of P5, we could not rule out that any of these two proteins or if a mixture of these two proteins could potentially form the penton base. Therefore, two initial models were generated. One model used the P31 protein sequence and the other used the P5 protein sequence as part of the input. In the initial observation of all the predicted segments of the two models, the P31 protein segment with residues 112–126 and the P5 protein segment with residues 107–121 occupied the same region of the sectioned map (*Figure 1E,F*). On closer inspection of the side chains from the two models and their fit into the CryoEM map densities in this region, the P5 protein side chains fit clearly at 3.2 RMSD and the P31 side chains could only be fitted at RMSD values lower than 2.6. As the CryoEM map was generated by an icosahedrally symmetrized reconstruction, we assumed that the densities of P5 and P31 were averaged. This hinted that the penton base could be a heteropentamer formed by both P5 and P31.

With the assumption that the penton base could be a heteropentamer, the P5 N-terminal domain (residues 1–124) and P31 (residues 1–126) were modeled using the icosahedrally averaged map at ~3.0 and ~2.2 RMSD respectively in *Coot* (*Emsley et al., 2010*) (*Figure 4E* and *Figure 4—figure supplement 3*). The prominence of the amino acid side chain densities varied based on the protein sequence conservation between P5 and P31 (*Figure 4—figure supplement 1*). All identities and most of the conserved substitutions in the sequence alignment resulted in clearer well-defined amino acid side chain densities and the non-conserved substitutions resulted in amino acid side chain densities that were averaged between the respective side chain densities (*Figure 1D–F* and *Figure 4—figure supplement 2*).

Another unexpected feature was observed at a lower RMSD of about 1.0, a branching of the density close to the N-terminal region of either P31 or P5. One prominent branch, which was earlier used for modeling and another branch that is only visible at lower RMSD. This suggested that the N-terminal region of either the P31 or P5 could have an alternate conformation. On inspecting the residues from both the P31 model and the N-terminal domain model of P5, that were close to the branch region, P31 was ruled out as a potential candidate. P31 has Val11-Thr10-Met9 residues, which could not be fitted into the branched density without severely distorting the Cα backbone. P5 in the same region has Ser9-Gly8-Gly7. The 2 Glycine residues provided the needed backbone (Cα chain) flexibility that could facilitate this 'special case' of N-terminal conformation (*Figure 4G* and *Figure 4—figure supplement 5*). In a typical arrangement, the N-terminal ends of both P5 and P31 would hug its neighboring subunit counter-clockwise and stabilize the penton complex (*Figure 4—figure supplement 7*). In the special case as described above, the N-terminal end of the P5 protein wedges itself between two adjacent P3 subunits (*Figure 4H*).

In the icosahedrally symmetrized reconstruction, at lower contour levels (0.065 in chimera), the map showed smeared densities above the 5-fold vertices of the viral particle (*Figure 4A–B*). This could be due to a symmetry mismatch of the proteins present in the region. Accordingly, the smeared region over the five-fold vertex was isolated and resolved by capsid signal subtraction followed by localized asymmetric (C1) reconstruction (see Materials and methods). All the classes generated by 3D classification showed a single protruding density except one of the classes, which revealed two significant densities; one poorly resolved knob-like density and another more well resolved density closely interacting with one of the monomers of the penton (*Figure 5—figure supplement 1*). On closer inspection of every 3D class generated during the process of localized asymmetric reconstruction, we noticed that three of the subunits of the penton base had a stem-like protrusion close to the 5-fold axis, extending outwards and interacting with each other forming a thick stalk (*Figure 4C–D*). The other two subunits lacked the stem-like protrusion. The class that revealed the two significant densities also showed that one of the densities interacted with the thick stalk (*Figure 5A–B*).

The number and arrangement of P5 and P31 forming the penton were confirmed by two independent methods, a localized asymmetric reconstruction and a focused classification to a resolution of 4.41 Å and 4.25 Å, respectively (*Figure 4—figure supplement 6*). All the classes from the 3D classification by localized reconstruction showed that only three subunits formed the stem-like protrusion that interacted with one another to form a thicker stalk. P31 terminates close to the fivefold and thus cannot form the stem-like protrusion whereas P5 residues (121-124) continue up and outward and these residues have the potential to form the stem-like protrusion (*Figure 4E* and *Figure 4—*

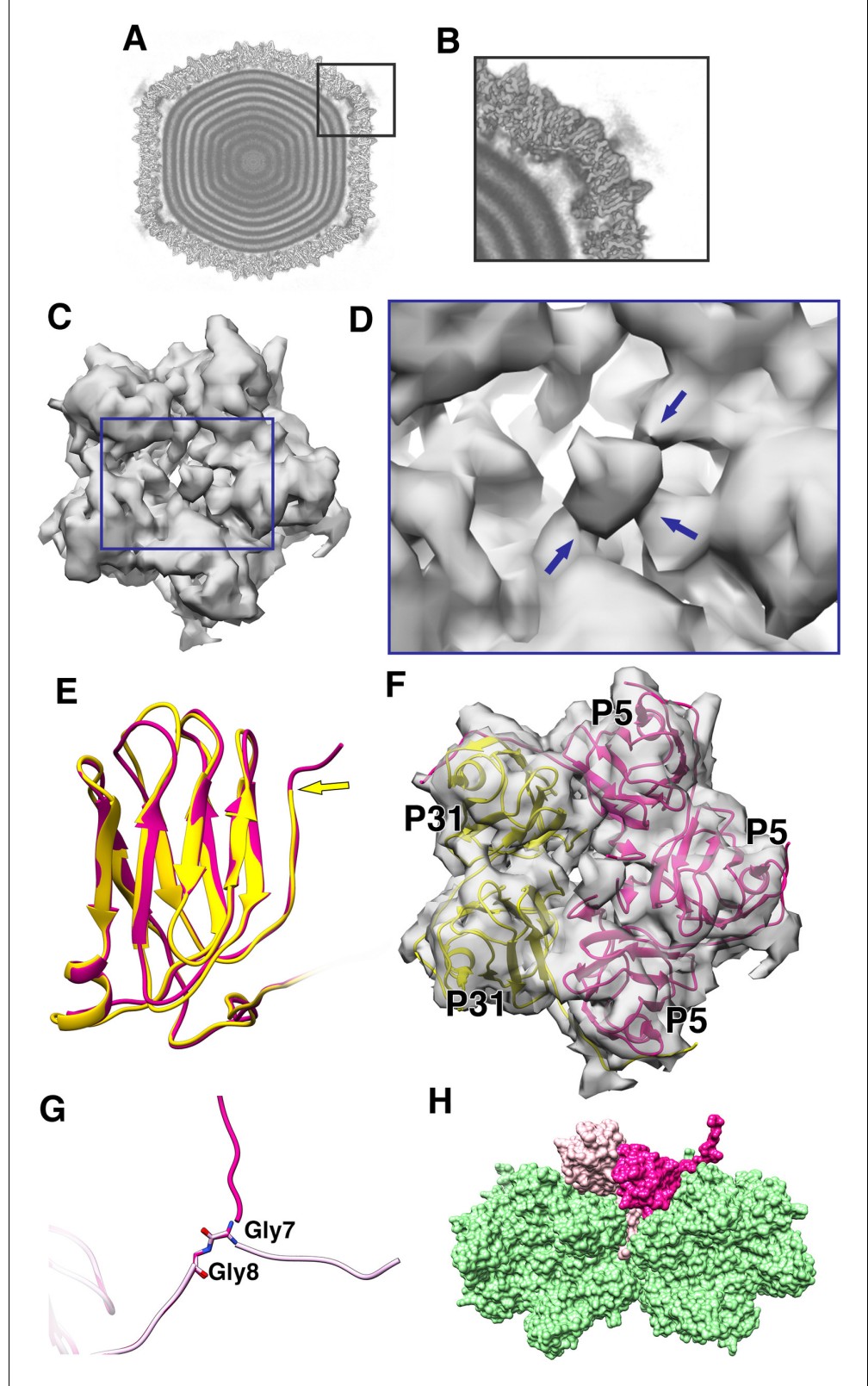

**Figure 4.** Penton as a heteropentamer of P5 and P31. (A) Icoshedrally averaged CryoEM map of PR772 showing the smeared densities at the five-fold vertices (Visualized using USCF Chimera, with volume viewer parameters: Style solid, step 1 and level 0.065, Plane 418, Axis Y, Depth 23). (B) Magnified image of a vertex showing the smearing of densities due to mismatch in symmetry. (C) Typical top view of the vertex map from the localized

*Figure 4 continued on next page*

*Figure 4 continued*

asymmetric reconstruction shown as a gray surface and the highlighted region is magnified in (D). In (D), the three stem-like protrusions (indicated by blue arrows) which interact with one another to form the stalk are shown. (E) The structure of P31 (yellow) and P5 (pink, residues 1–124) are superimposed and P31 terminates close to the five-fold (indicated by a yellow arrow) and P5 (residues 121–124) continues upward. (F) The N-terminal domain of P5 and P31 are fitted into the localized reconstruction vertex map as rigid bodies and the P5 residues 121–124 fit into the stem-like density shown in (D). (G) Shows Gly7 and Gly8 residues at the N-terminal end of the P5 subunit and the two conformations (regular conformation as a bright pink structure and the special conformation as a pale pink structure). (H) Shows how the two conformations of the N-terminal region of P5 interact with the neighboring protein. The regular confirmation of the N-terminal end of P5 (bright pink) hugs the neighboring penton subunit whereas the special conformation (pale pink) is wedged in-between the P3 trimers (green).

DOI: https://doi.org/10.7554/eLife.48496.011

The following figure supplements are available for figure 4:

**Figure supplement 1.** Shows the protein sequence conservation between P31 and P5.
DOI: https://doi.org/10.7554/eLife.48496.012

**Figure supplement 2.** Comparison of model fitted into map density in few of the conserved regions between P31 and P5.
DOI: https://doi.org/10.7554/eLife.48496.013

**Figure supplement 3.** Modelled P31 protein and its map density fit.
DOI: https://doi.org/10.7554/eLife.48496.014

**Figure supplement 4.** Comparison of residues from the N-terminal domain of P5 that contribute to the formation of the stalk and the lack of residues in P31 that contribute to the density.
DOI: https://doi.org/10.7554/eLife.48496.015

**Figure supplement 5.** Two conformations of the P5 N-terminal base.
DOI: https://doi.org/10.7554/eLife.48496.016

**Figure supplement 6.** Reconstruction of the penton region.
DOI: https://doi.org/10.7554/eLife.48496.017

**Figure supplement 7.** P31 and P5 residues stabilizing the heteropentameric penton.
DOI: https://doi.org/10.7554/eLife.48496.018

*figure supplement 4*). Three copies of the model of the P5 N-terminal base were fitted into the penton density from the localized asymmetric reconstruction. These models did indeed fit the density (*Figure 4F*). The overall map:model correlation reduced when the models of P31 and P5 N-terminal domain were swapped with each other in the localized reconstruction and the focused classification density maps. The orientation and alignment of P5 residues (121-124) also confirm previous predictions, which showed the formation of a triple helix with a collagen-like motif (residues 124–140) (*Huiskonen et al., 2007*; *Caldentey et al., 2000*). The poorly resolved knob-like density represents the trimerized C-terminal domain of P5 (*Caldentey et al., 2000*; *Merckel et al., 2005*). By this it can be concluded that 3 copies of P5 and 2 copies of P31 form the penton base in PR772 (*Figure 4F*).

## P2 monomer is bound to P5 and stabilized by the P5 stalk

With the localized asymmetric reconstruction and focused classification methods, the structure and composition of the penton base was established. It is now known that the stalk-like density observed, emanating from the penton base, is built with three copies of the P5 protein, forming a collagen-like motif and eventually the unresolved knob domain.

As described previously, one of the 3D classes, with 467,631 sub-particles from a total of 3,340,224 sub-particles, also revealed an extra rigid density closely interacting with the penton base and the protruding stalk (*Figure 5A–C* and *Figure 5—figure supplement 1*). In previous studies, it was shown that P2 and P5 could be interacting at the vertex (*Bamford and Bamford, 2000*). The crystal structure of P2 (PDB:1N7U) from PRD1 (98% protein sequence identity to PR772) was fitted into the density. The fit was good and the density that was resolved by localized reconstruction was sufficient to fit the domains I and II, forming the head and the domain III, forming the tail of P2. The off-centred arrangement of the tail with respect to the head allowed us to specifically assign the map density to the respective regions of the P2 protein (*Figure 5A–C*).

The interaction of P5 with P2 can be compared to a ball and socket joint, for instance as seen in a human shoulder, where the N-terminal base of P5 is the ball with domain I and domain II of P2 acting

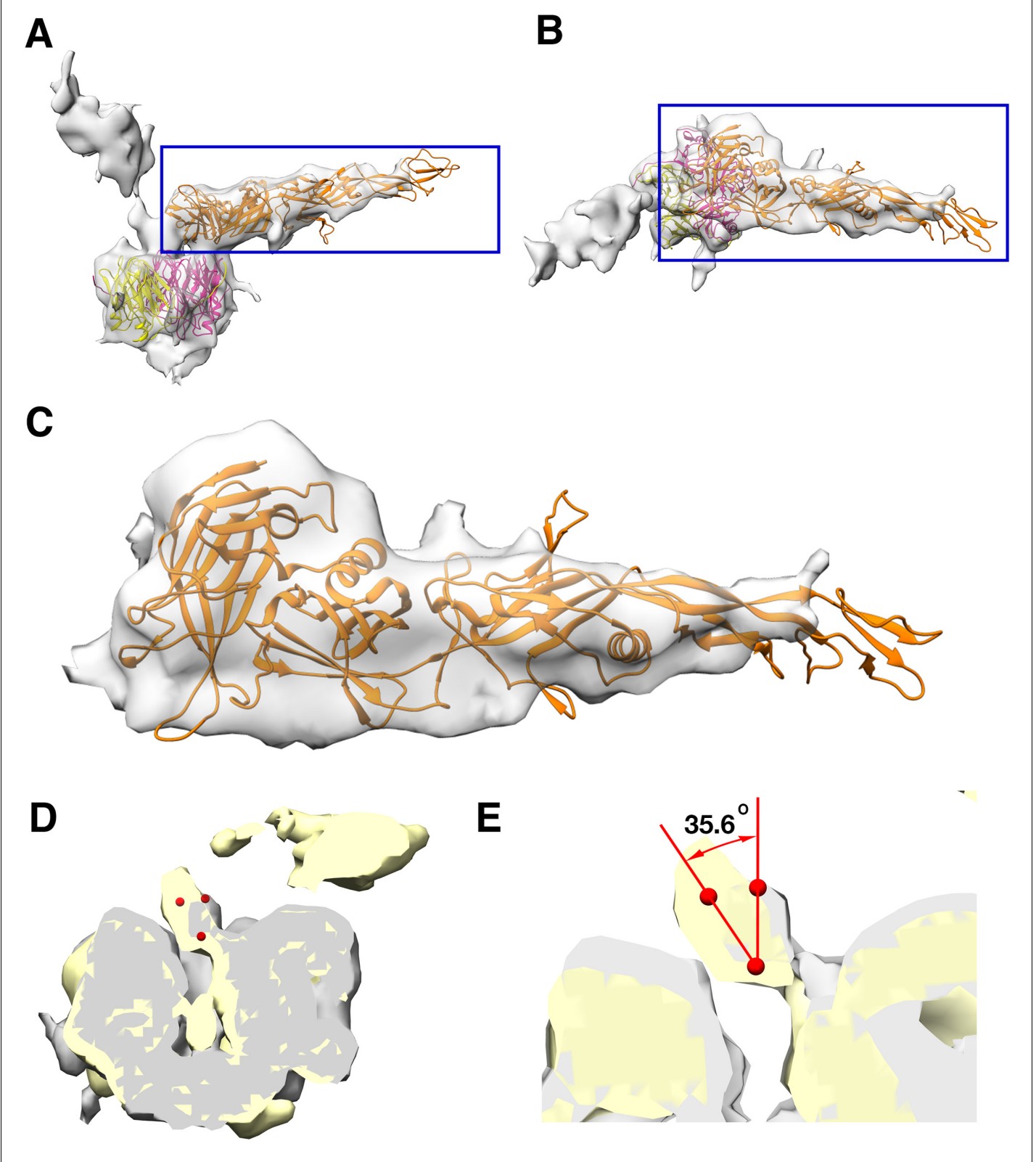

**Figure 5.** Monomeric P2 bound to P5. Localized asymmetric reconstruction of the vertex complex showing the two protruding densities as gray surfaces where (A) is the side view and (B) is the top view. P2 (orange), P5 N-terminal base (bright pink) and P31 (yellow) structures fitted into these map densities. (C) Isolated CryoEM density that represents the P2 subunit. It represents the region highlighted by the blue box in (A) and (B). (D and E) Superimposed vertex maps with P2 bound (yellow) and without P2 bound (gray). In classes where P2 is bound to P5, the stalk region is nudged by

*Figure 5 continued*

$\approx 35.6^{0}$ when compared to the classes where P2 is not bound to P5. Chimera: Volume Trace tool was used to place the red spheres in the density and Chimera: Measure Angles tool to determine the angles.

DOI: https://doi.org/10.7554/eLife.48496.019

The following video and figure supplements are available for figure 5:

**Figure supplement 1.** Maps generated by localized reconstruction of the sub-particles.

DOI: https://doi.org/10.7554/eLife.48496.020

**Figure supplement 2.** Distribution of electrostatic potential and hydrophobic surface in P2 and P5.

DOI: https://doi.org/10.7554/eLife.48496.021

**Figure 5— video 1.** Shows the distribution of coulombic electrostatic potential on the surface of P2 and P5 and the regions of interaction between the N-terminal domain of P5 and P2.

DOI: https://doi.org/10.7554/eLife.48496.022

as a socket (*Figure 5—video 1*). The estimation of the electrostatic surface potential of both P2 and P5 shows good charge complementarity in and around the regions of interaction. These regions also have similar hydrophobicity (Kyte-Doolittle scale) (*Figure 5—figure supplement 2* and *Figure 5— video 1*). P2 also interacts with the collagen-like motif of the P5 stalk, thus introducing some rigidity to the C-terminal of the P5 trimer. The presence of P2 nudges the P5 stalk by ~$36^{O}$ from its usual position (*Figure 5D–E*). The P5 stalk acts as a linchpin that locks the P2 molecule and restricts it from swivelling around the 5-fold vertex, thus stabilizing the complex.

The occupancy of P2 on the 5-fold vertices appears to be significantly lower in bacteriophage PR772 than what was observed from other members of the *Tectiviridae* family. The 3D classification of the signal subtracted and isolated vertices revealed that only 467,631 sub-particles from a total of 3,340,224 extracted sub-particles, showed the presence of a density representing P2. This accounts for 16% of all the sub-particles used or about two vertices in an intact viral particle, assuming an equal distribution among the viral particles.

## Overall architecture of PR772

P30 adopts an extended conformation in an intact viral particle and it is found wedged in between the trisymmetrons (*Figure 2A*, *Figure 6—figure supplement 2*). Two copies of P30, interlocked at the N-terminal hook, span between the adjacent vertices. In an intact viral particle, P30 forms a cage-like structure, which stabilizes the trisymmetrons themselves and also interact with the neighboring trisymmetrons to form a closed network (*Figure 2A* and *Figure 6—figure supplement 2*). The residues Tyr62 – Ile64 and Val32 – Arg35 of P30 form beta sheets with residues Thr384 – Leu386 from subunit c of P3 trimer one and residues Thr384 – Asn388 from subunit c of P3 trimer two respectively (*Figure 2A* highlighted regions). These C-terminal regions of P3 are sandwiched between P30 and the lipid membrane (*Figure 2A* highlighted regions). P30 is also sandwiched between two adjacent P3 subunits of the trimer from a neighboring trisymmetron (*Figures 2A* and *6B* and *Figure 6—video 1*).

Five copies of P16 bridge the penton to the five trisymmetrons that meet at the vertex complex. The structure of P16 was only partially resolved. The long C-terminal trans-membrane helix of P16 (Leu7 - Ala28) was poorly resolved and barely visible at 0.6 RMSD. The high-resolution structure of the disorganized region (Tyr58 - Ile96) of P16 also evades us. It was poorly resolved with no visible side chains and part of the density map representing the Cα backbone was missing even at very low RMSD. Due to poor signal to noise ratios, these regions were not modeled, but the difference density map of the vertex region and the modeled penton showed that the C-alpha backbone continued into the empty pocket beneath the penton and also interacts with the penton close to the 5-fold axes (*Figure 6—figure supplement 3*). However, the partial model shows that P16 forms a clamp-like complex using the loop Asn44 -Val55 and helix Asn101 - Ala115, which attaches onto the loop Val242 - Tyr247 of one of the P3 subunits of trimer one and locks it with the adjacent P3 subunit of trimer one from the neighboring trisymmetrons (*Figure 6B*). This further locks the P3-P30-P3 sandwich and makes it stable. As was mentioned earlier, to accommodate the P30 protein in the P3-P30-P16 complex (*Figure 6* and *Figure 6—video 1*), the loop formed by residues Tyr351-Val358 of the P3 subunit in trimer one is flipped compared to the orientation seen in this loop of the P3 subunit with a N-terminal long helix (*Figure 2—figure supplement 3D–F*).

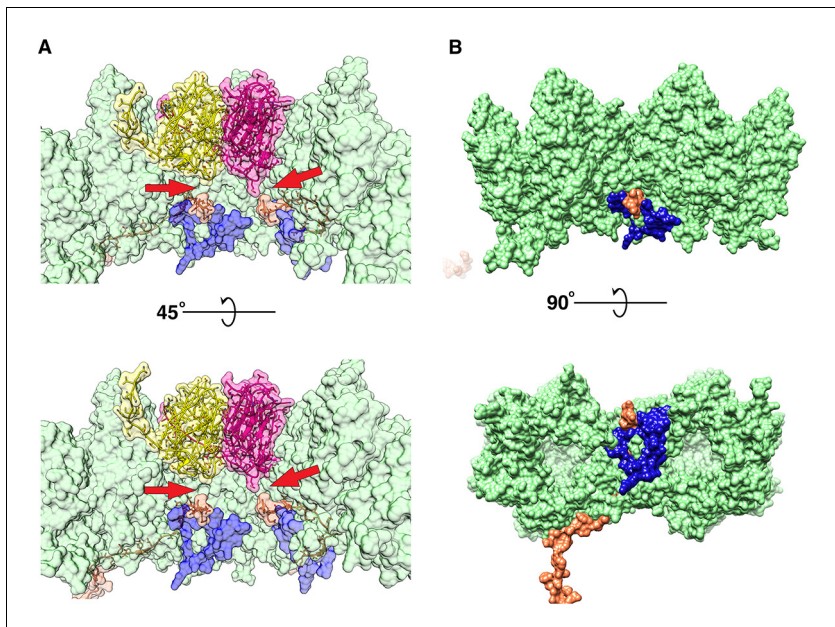

**Figure 6.** P16 and P30 intractions close to the vertex. (**A**) Shows the difference in interaction (pointed by red arrow) of the C-terminal Gly84 of P30 (orange) with P5 (bright pink), P31 (yellow) and P3 (green). The hydrophobic intraction of P30 with P5 is more obvious. (**B**) Shows the P3-P30-P16 complex (the view is similar to (**A**) but one copy of P16 and P30 are shown and the penton proteins are hidden), P16 (blue) locks the two adjacent P3 trimers (trimer 1)(green) and the P30 protein (orange).

DOI: https://doi.org/10.7554/eLife.48496.023

The following video and figure supplements are available for figure 6:

**Figure supplement 1.** Partial model of P16 and its map density fit.
DOI: https://doi.org/10.7554/eLife.48496.024
**Figure supplement 2.** P30 network.
DOI: https://doi.org/10.7554/eLife.48496.025
**Figure supplement 3.** Difference density map showing the disordered region under the penton.
DOI: https://doi.org/10.7554/eLife.48496.026
**Figure supplement 4.** Schematic of the infection mechanism model of bacteriophage PR772.
DOI: https://doi.org/10.7554/eLife.48496.027
**Figure 6— video 1.** Movie showing the P3-P30-P16 complex.
DOI: https://doi.org/10.7554/eLife.48496.028

At 2.0 RMSD, we noticed that the density for the C-terminal Gly84 of P30 was pointing towards the penton base and away from the inner lipid membrane. This is different from what is seen in bacteriophage PRD1 (*Abrescia et al., 2004*). In case of P5, the C-terminal Gly84 of P30 seems to interact with the Met19 residue of a P5 subunit by a hydrophobic interaction (*Figure 6A*). With P31, the hydrophobic interaction is not evident.

Three copies of P5 and two copies of P31 are held together by their N-terminal residues. Met9 – Val14 of P31 and Ser9 – Tyr13 of P5 inter-digit with the neighboring subunit of the pentamer as β-sheets. Residues Gln8 – Asn2 and Gly8 – Met1 of P31 and P5 respectively continue further and hug the neighboring subunits (*Figure 4—figure supplement 7*). As was mentioned earlier, residues Gly8 - Met1 of P5 can in special cases have an alternate conformation where they are wedged in between the adjacent P3 subunits (*Figure 4H*). There is no obvious direct interaction between the penton and the P3 subunits of the capsid except for the special cases mentioned above. The disorganised region of P16 seems to interact with the penton base and helps in binding the vertex complex to the trisymmetrons (*Figure 6—figure supplement 3*).

## Discussion

Each trisymmetron of PR772 is composed of 12 trimers of the P3 protein that are in turn bound by P30, an overall arrangement similar to that of the close relative PRD1 (*Abrescia et al., 2004*). The high-resolution structure of PR772 confirms the previously shown variability of the N and C-terminals of the P3 protein depending on the locations within the trisymmetron. The high-resolution also allows us to extend the analysis of these conformations and show that the N-terminal of P3 can adopt three different conformations, not two as previously described. (*Abrescia et al., 2004*; *Butcher et al., 2012*) (*Figures 2A* and *3B*). The newly discovered conformation of the N-terminal region of P3, a long helix with a kink, is structurally critical to accommodate the interlocking region formed by the N-terminal hooks of P30 during the initiation of particle assembly. The N-terminal region of P3 shows plasticity and play an important role in both stabilizing the trimer and anchoring the trimer complex to the membrane. The C-terminal region of P3 also shows differences in its conformation as compared to PRD1. In trimer 1 (*Figure 2A,C and H*) subunit a and subunit c have elongated C-terminal regions that interact with the adjacent P3 subunits of the neighboring trisymmetrons and in-turn locking on to P30 (*Figure 2* right hand side insert; and *Figure 6*). A similar arrangement can be seen in the C-terminal region from subunit c of trimer two and subunit a of trimer 4 of the neighboring trisymmetrons (*Figure 2* left hand side insert). The presence of P16 close to the penton region makes the interaction of all the neighboring trimer one more stable (*Figure 6*)

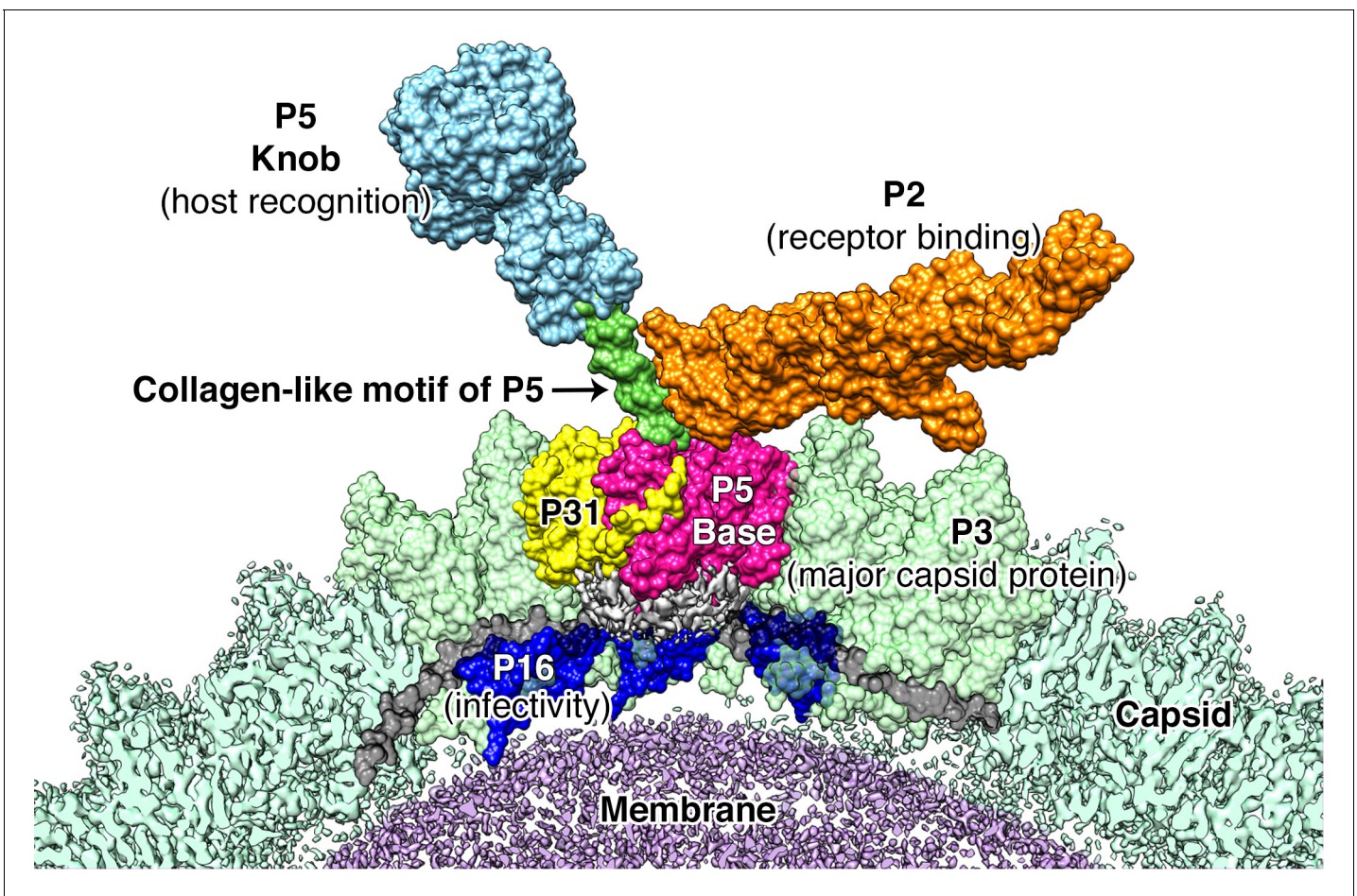

**Figure 7.** Proposed model for the vertex region. Shows the structural arrangement of P5 (P5 N-terminal base shown in bright pink, Collagen-like motif of P5 shown in bright green and P5 Knob is shown in pale blue), P31 (yellow) and P2 (orange) that form the vertex. The region below the vertex complex (shown as a gray density) is the unorganized/unmodeled region that links the surrounding P16 (blue) and the P30 (black). Capsid is shown in pale green (raw CryoEM map is labeled as Capsid and the modeled region is labeled as P3). The lipid membrane is shown in purple.
DOI: https://doi.org/10.7554/eLife.48496.029

and also anchors the whole vertex complex to the membrane (*Figure 7*), emphasizing the role of P30 and P16 in maintaining the size and structure of the icosahedral viral capsid.

Our studies show that unlike what was predicted for PRD1, the penton base of PR772 is an asymmetric heteropentamer consisting of three copies of P5 and two copies of P31 (*Figure 4F*). P31 has high sequence similarity with the N-terminal domain of P5. Previously, it was shown that P31 can replace P5 to form the penton and produce an intact viral particle (*Bamford and Bamford, 2000*). Even though the viral particles were intact, they were non-infectious due to the lack of the viral receptor-binding protein, P2, that is bound to the host recognition protein, P5 (*Bamford and Bamford, 2000*; *Grahn et al., 1999*). In another study with the PRD1 sus525 mutant that lacks P31, it was also shown that these particles lacked the vertex complex (*Rydman et al., 1999*). With the current model, P31 would promote the formation of an intact viral particle by avoiding the steric-hinderances that could occur during the formation of the vertex complex, if the P31 subunits were replaced by P5. A penton with 3 copies of the P5 subunits will support (i) the formation of a stable collagen-like motif (*Figure 7*) and (ii) the formation of the C-terminal host recognition domain of P5. A heteropentameric base of the vertex complex with three copies of P5 and two copies of P31 address both these issues.

The members of the *Tectiviridae* family have been shown to have structural similarities with adenoviruses despite infecting different hosts. In adenoviruses, the symmetry mismatch seen in the vertex region is solved by the interaction of the tail region of the trimeric fibre with three of the five grooves formed by the subunits of the penton base (*Cao et al., 2012*). Unlike in adenoviruses, a similar problem in PR772 is solved by domain swapping. The trimeric P5 protein responsible for the formation of the spike, swaps the N-terminal base domain with three copies of the P31 protein (*Figure 4F*).

The localized asymmetric reconstruction shows that P2 is bound to P5 (*Figure 5*). In the current CryoEM data, the orientation of P2 with respect to the penton base is reversed when compared to what was described for PRD1 (*Huiskonen et al., 2007*; *Xu et al., 2003*). Here, the beta-propeller motif of P2 with domain I and II, interacts with the N-terminal base of P5 and also with the stalk. This was also speculated on in a previous study, where it was noted that members of the *Tectiviridae* family with large sequence variation in the beta-propeller motif of P2 also showed variations in the P5 subunit as a compensatory effect (*Saren et al., 2005*). P2 interacts with both the N-terminal base and the stalk region of P5 but does not interact directly with any other structural protein in PR772. The interaction of P2 with the P5 stalk stabilizes the C-terminal region of P5 (*Figure 5*). In the case of PRD1, P2 seems to occupy all vertices except for the unique packaging vertex (*Butcher et al., 2012*; *Caldentey et al., 2000*; *Saren et al., 2005*). In this scenario, the special conformation in which the N-terminal residues of P5 are wedged in between the neighboring P3 subunits was found to be prominent (*Abrescia et al., 2004*). In the current model of PR772, P2 seems to occupy only 2 vertices of the icosahedral shell. The lower quantities of P2 in PR772 were also indicated in a previous study, comparing the abundance of different viral proteins from various members of the *Tectiviridae* family using western blots (*Saren et al., 2005*). In PRD1, it was shown that the loss of P2 from a packed viral particle, resulted in spontaneous release of genomic DNA, leading to loss of infectivity and the formation of empty particles (*Grahn et al., 1999*). In a previous study of PR772, it was shown that after purification, the concentration estimates of the particle by plaque assay and nanoparticle tracking analysis were similar (*Reddy et al., 2017*). PR772 particles remained intact and infectious after purification. The intact PR772 particles can also be observed in the CryoEM micrographs (*Supplementary file 3*). So, it is unlike that P2 was lost during the purification.

In the icosahedrally averaged model of PR772, we see that the special conformation of P5 where the N-terminal residues are wedged between the P3 subunits of the adjacent trimer one is not prominent, but these P5 densities are visible at lower RMSD values. The presence of P2 in the vertex complex may coincide with the N-terminal region of P5 adopting the special conformation. It was reported that, compared to the CryoEM map of a wild type PRD1 virion, the CryoEM map of *sus690*, a mutant PRD1 virion that lacks P2 and P5, showed absence of density in the region where we see the N-terminal of P5 wedged between two neighboring P3 subunits. The lack of this density was attributed to the conformational changes in P31 due to the binding of P5 or the presence of P5 in between P31 and the P3 trimer (*Huiskonen et al., 2007*). In light of the current findings in PR772, P5 is known to be part of the heteropentameric base forming the penton and the absence of the above-mentioned density confirms that P5 adopts the special N-terminal conformation and not P31.

The above mentioned observations and the quantitative analysis of different proteins from PRD1 and PR772 using western blots (*Saren et al., 2005*) could explain the discrepancy in P2 packing in PR772. As mentioned earlier, the presence of P2 in the vertex complex may coincide with the N-terminal region of P5 adopting the special conformation. This special conformation of N-terminal region of P5 could be linked to the low copy number of the membrane protein P14 (or higher copy number of P7) observed in PR772 when compared to PRD1 (*Saren et al., 2005*). In PRD1, gene VII codes for P7 and the 3' region of the same gene also codes for P14. P14 is smaller and it consists only of the membrane anchor domain and lacks the trans-glycosylase domain that is seen in P7. These proteins, P7 and P14, are proposed to forms a hetero-multimeric complex located under the vertices of the PRD1 particle (*Mattila et al., 2015*; *Rydman and Bamford, 2000*). In an intact PRD1 particle, the copy numbers of P7 and P14 are similar but in case of PR772, the copy number of P14 is significantly lower and the copy number of P7 seems higher (*Mattila et al., 2015*; *Saren et al., 2005*). In PR772, the copy number of P14, the occurrence of the special conformation of the N-terminal region of P5 protein and the number of P2 proteins packed in an intact viral particle seems to be related. In PR772, the presence of the larger P7 protein in higher numbers at the vertex (along with other proteins in the disordered region) might hinder the formation of the special conformation of the N-terminal region of P5, which could reduce the number of P2 proteins packed in the particle.

The penton has no obvious direct interaction with the neighboring P3 subunits of the capsid, except in the rarely occurring special conformation of P5 where the N-terminus is wedged in between the P3 subunits of trimer one as mentioned above (*Figure 4G–H*). The difference density map of the vertex region and the modeled penton shows that the disordered region of P16 interacts with the penton (*Figure 6—figure supplement 3*). These interactions of P16 with P3, P5, P31 along with the membrane seem to play a central role in the formation of the vertex complex and anchoring it to the membrane. In earlier Raman spectroscopy studies on PRD1, it was shown that in the initial stages of viral assembly, P3 and P5 formed a precursor shell assisted by assembly factors and other membrane-associated proteins such as P16 (*Bamford et al., 1990*; *Bamford et al., 1995*). Similarly, in PR772 the P5 trimers could be accommodated during the formation of the procapsid and two copies of P31 are later added to the penton region, forming a heteropentameric penton.

From the localized asymmetric reconstruction of the vertex complex, one can notice that there is a variation in the map density below the penton. The map density below the P5 subunits is significantly different when compared to the same region below the P31 subunits. This suggests that, along with the interactions of P16 with P5 and P31, there could also be poorly resolved proteins like P11, P7, P14 or P18, etc; in the region (*Luo et al., 1993*; *Mattila et al., 2015*; *Rydman and Bamford, 2000*). The variation in the map density in this region is also compounded by other interactions, like that of the C-terminal Gly84 of P30 with the Met19 of P5 subunits and lack thereof with respect to the P31 subunits.

Preliminary results for the whole particle asymmetric reconstruction using symmetry relaxation in EMAN2 of the wild type PR772 do not show the presence of a unique packaging portal in the dormant particle in contrast to PRD1. All the vertices of PR772 show the heteropentameric arrangement of the penton with three subunits showing the stalk and the other two without the stalk (*Figure 4C–D*, *Supplementary file 2*).

*Figure 7* shows our model of the vertex complex in PR772. Using this model, we propose a mechanism for the initiation of infection. In analogy to studies on PRD1, it is known that P5 is needed for host recognition, but the binding of P5 to the host is transient (*Grahn et al., 1999*). The host binding is stabilized by the high-affinity interaction of P2 and its receptor, locking the viral particle to the host (*Grahn et al., 1999*). The relative changes between P5 and P2 upon binding to the receptor on the host could trigger the disruption of the vertex complex by pulling the N-terminal region of the P5 trimers that are wedged between the P3 subunits. This will in turn disrupt the interaction between the P5 proteins and P30 as seen in *Figure 6*. This disruption cascades further and disturbs the interaction between P30 and P16, rendering the whole vertex complex unstable. P30 dimers could also transduce these effects to the neighboring vertices. The disruption of the vertex complex exposes the viral membrane and membranous proteins like P18, P11/P7, P32 etc; to the host surface to facilitate the formation of the membranous tube for DNA transport. P16, that holds the vertex complex, along with other membranous proteins, could act as a protein tether that would help in moving the lipid membrane closer to the host (*Figure 6—figure supplement 4*).

## Materials and methods

### Preparation and purification of PR772

Bacteriophage PR772 (ATCC BAA-769-B1) was propagated on *Escherichia coli* K12 J53-1. It was purified as previously described (*Reddy et al., 2017*) and further concentrated to facilitate testing various concentrations of the viral sample during grid optimization for CryoEM. This method yielded about 2–4 mL of viral particles with a concentration of 1 mg/mL using 10–20 agar plates. The sample was further concentrated to 20 mg/mL by using an ultracentrifuge. The sample was added into an ultracentrifuge tube and then a solution of Caesium Chloride in buffer (HEPES 20 mM, NaCl 100 mM, $MgSO_4$ 1 mM, EDTA 1 mM, pH 8.0) at a density of 1.34 g/mL (g/cm$^{-3}$) was gently layered on top. The mixture was centrifuged at 100,000 × g for 30 mins. The intact viral particles migrated to the top as a fine band and the broken particles along with any free DNA that was released from the broken particles stayed at the bottom of the tube. The top band with the intact viral particles was extracted using a needle and syringe. The concentrated sample was dialyzed over-night with the above-mentioned buffer to remove caesium chloride.

### CryoEM grid preparation and data collection

The condition for CryoEM grid preparation was optimized for collecting a large number of particle images. For vitrification of the viral sample by plunge freezing into liquid ethane, we used a Vitrobot Mark IV (ThermoFisher). The best grid condition with uniform sample distribution was obtained by applying 3 µL of 7 mg/mL concentrated viral sample solution on a glow-discharged C-Flat grid CF-2/2–2C under 100% humidity at room temperature.

The data were collected on a Titan KRIOS (ThermoFisher) equipped with a K2 Summit (Gatan) direct electron detector and a GIF Quantum LS (Gatan) energy filter. All the data were collected at a magnification of 130 k in EFTEM mode with a pixel size of 1.06 Å. The slit width of the energy filter was 20 eV. The dose rate was 4.4 e$^-$ per Å$^2$ per second with a total exposure of 9 s resulting in a total dose of ~40 e$^-$/Å$^2$. The total dose was distributed over 40 frames in each movie. 3220 movies were collected.

### Image processing

The movie frames were corrected for beam induced sample motion and aligned using MotionCor2 (*Zheng et al., 2017*). The first 3 frames of the movies were skipped and the rest were aligned. These aligned frames were averaged with and without dose weighting. The non-dose weighted image stacks were used to estimate defocus and correct CTF using CTFFIND4 (*Rohou and Grigorieff, 2015*). All estimated fits of defocus and CTF were visually inspected. All images with significant astigmatism or a prominent ring due to crystalline ice around 3–4 Å were discarded.

### Whole particle reconstruction

A total of ~3200 images were used to auto pick 56275 particles using template matching in RELION (*Scheres, 2012*) (version 2.1 beta 1) (*Kimanius et al., 2016*). The 2D classes generated by 2D classification of 710 manually picked particles were used as templates for auto-picking. The auto-picked particles were binned 2 × during the extraction (box size of 429 × 429 and 2.12 Å/pix). Extensive reference free 2D classification was performed to remove any particle images with ethane contaminants or broken/empty viral particles. The classes with good 2D averages were selected and the particles from these classes were extracted. This resulted in 51893 particles that were used for 3D classification. RELION: 3D initial model tool, which is based on stochastic gradient descent, was used to generate an ab-initio reference map for 3D classification. The 2 × binned particle images were used to generate a low-resolution icosahedrally averaged map. This low-resolution map was used as a reference for 3D classification. The 3D classification was performed with icosahedral symmetry (I4) applied. The most dominant class, with 46348 particles, was selected and the particles were extracted for final refinement. Icosahedral symmetry (I4) was also applied during the final refinement. The refinement with the 2 × binned particles reached Nyquist sampling (~4.24 Å here). The particles from the final iteration step of the refinement were re-extracted without binning and further refined. The reference map was also scaled to match the new box and pixel size (858 × 858

and 1.06 Å/pix respectively) using e2proc3d.py from EMAN2.1 package (*Bell et al., 2016*) (*Supplementary file 3*).

After refinement, the maps were corrected for Ewald sphere effects using RELION 3.0 beta 2. These maps were post-processed. A soft binary mask was generated using the 15 Å low pass filtered map extended by 10 pixels and with a soft edge of 15 pixels. An initial binarization threshold of 0.001 was used to include all the map features (i.e, internal membrane, etc) in the mask. This was used as a solvent mask during post processing. The map was corrected for the detector's Modulation Transfer Function (MTF) and sharpened with an inverse B-factor. Overall gold standard FSC@0.143 was estimated using two independently processed half maps. ResMap (*Kucukelbir et al., 2014*) was used to estimate the variation of resolution across the two unfiltered half maps.

## Localized asymmetric reconstruction of the five-fold vertex

The vertex complex was reconstructed using 2x binned particles by localized reconstruction (*Ilca et al., 2015*). Particle images from one half map were symmetry expanded (icosahedral to C1) and signal subtracted to remove the signal of the viral capsid and the genome from the particle images. Sub-particles representing each of the vertex region, with a box size of $100 \times 100$ pixels were extracted from the signal subtracted particles. This produced about 3,340,224 sub-particles. These particles were 3D classified without imposing symmetry and disabling image re-alignment. An initial reference map was produced by back projecting the extracted sub-particles using the previously calculated Euler angles. This initial reference map was low pass filtered to 45 Å and used as a reference map during 3D classification. The resolution of the expectation step was limited to 10 Å and a spherical mask of diameter 200 Å was added using the mask diameter and flatten solvent options provided by RELION to reduce the effects of systematic noise due to signal subtraction.

The classes generated from the 3D classification were analyzed and the particles from the class showing two protruding densities were selected and the 3D classification was now further refined with parameters similar to what was mentioned above but with the local image re-alignment enabled. The part of the map density that we were interested in was partially clipped due to the spherical mask that was used. So, the sub-particles were re-extracted with a larger box size of $140 \times 140$ pixels. These newly extracted particles with a larger box were back projected using the Euler angles calculated during the previous 3D classification.

To improve the resolution of the penton base region, another set of 3D classification was performed without limiting the e-step to 10 Å using the same set of sub-particles but applying a soft-mask around the penton.

## Focused classification of the Penton base

To validate the penton map generated from the localized reconstruction, we also applied a different method, focused classification using the 2x binned particles. The particle representing the final converged map were symmetry expanded using *relion_particle_symmetry_expand* command. Similar to the localized reconstruction, signal subtraction was performed to remove the signal of the viral capsid and the genome from the particle images using *relion_project* command. We used the final converged map (without post-processing) from the icosahedrally averaged reconstruction to generate projections for signal subtraction. 3D classification with the signal subtracted particles was performed without imposing symmetry and disabling image re-alignment. The penton region extracted from the icosahedrally averaged map was low pass filtered to 45 Å and used as a reference map. A soft mask, around the penton region, was generated by extending the binary map of the penton region extracted from the icosahedrally averaged map by three pixels with a soft edge of 4 pixels. This was used as a focus mask to constrain the 3D classification to this region.

## Model building and refinement

Modeling the whole particle was difficult owing to the large size of the particle and the limited RAM available. The whole viral map was sectioned using the sub-region selection option in Chimera. The sectioned maps were optimized by local sharpening using PHENIX: Autosharpen (*Terwilliger et al., 2018*). These sectioned maps were used to generate a crude model using PHENIX: find helix and loops (*Adams et al., 2010*). The crude model was further used for de-novo modeling of the proteins

in *Coot* (*Emsley et al., 2010*). These models were further refined using PHENIX: Real space refinement to improve the model. All the models were individually refined and put together to form the asymmetric model. The model of the asymmetric unit was further refined against a new map encasing the asymmetric unit using PHENIX-Real space refinement. The model of the asymmetric unit was validated with *MolProbity* (*Chen et al., 2010*), Mtriage (*Afonine et al., 2018*) (*Supplementary file 4*) and EMRinger (*Barad et al., 2015*).

## Data availability

CryoEM Density maps and atomic models that support the findings of this study have been deposited in the Electron Microscopy Database and the Protein Databank with the accession codes EMD-4461 (Whole particle reconstruction), EMD-4462 (Vertex Complex), EMD-10237 (Localized reconstruction of the penton region), EMD-10238 (Focused Classification of the penton region) and PDB ID 6Q5U (Atomic model of the asymmetric unit).

## Acknowledgements

We would like to thank Kazuyoshi Murata and Naoyuki Miyazaki from National Institute for Physiological Sciences, Japan; for providing us the sample cryoem data set to test the feasibility of the project. We would like to thank Sjors Scheres from MRC Laboratory of Molecular Biology, UK and Björn Forsberg from Stockholm University, Sweden for their valuable feedback in handling large CryoEM maps in RELION.

## Additional information

### Funding

| Funder | Grant reference number | Author |
|---|---|---|
| Vetenskapsrådet | 828-2012-108 | Janos Hajdu |
| Vetenskapsrådet | 628-2008-1109 | Janos Hajdu |
| Vetenskapsrådet | 822-2010-6157 | Janos Hajdu |
| Vetenskapsrådet | 822-2012-5260 | Janos Hajdu |
| Knut och Alice Wallenbergs Stiftelse | KAW-2011.081 | Janos Hajdu |
| European Research Council | ERC-291602 | Janos Hajdu |
| Vetenskapsrådet | 349-2011-6488 | Janos Hajdu |
| Vetenskapsrådet | 2015-06107 | Janos Hajdu |
| European Commission | CZ.02.1.01/0.0/0.0/15_003/0000447 | Janos Hajdu |

The funders had no role in study design, data collection and interpretation, or the decision to submit the work for publication.

### Author contributions

Hemanth KN Reddy, Conceptualization, Formal analysis, Validation, Investigation, Visualization, Methodology, Writing—original draft, Writing—review and editing; Marta Carroni, Resources, Data curation, Validation, Writing—review and editing; Janos Hajdu, Resources, Supervision, Funding acquisition, Project administration, Writing—review and editing; Martin Svenda, Conceptualization, Resources, Supervision, Validation, Investigation, Project administration, Writing—review and editing

### Author ORCIDs

Hemanth KN Reddy https://orcid.org/0000-0002-4698-8005
Marta Carroni http://orcid.org/0000-0002-7697-6427

Janos Hajdu (iD) https://orcid.org/0000-0002-3747-2760
Martin Svenda (iD) https://orcid.org/0000-0003-1162-8285

**Decision letter and Author response**
Decision letter https://doi.org/10.7554/eLife.48496.046
Author response https://doi.org/10.7554/eLife.48496.047

## Additional files

### Supplementary files

• Supplementary file 1. Protein traces in the PR772 model and comparison of RMSD of different protein subunits from PR772 and PRD1, from a similar region of the map using Chimera and SuperPose.
DOI: https://doi.org/10.7554/eLife.48496.030

• Supplementary file 2. Preliminary results for the whole particle asymmetric reconstruction using symmetry relaxation in EMAN2 of the wild type PR772.
DOI: https://doi.org/10.7554/eLife.48496.031

• Supplementary file 3. Flowchart of the 3D reconstruction of the icosahadrally averaged PR772 map.
DOI: https://doi.org/10.7554/eLife.48496.032

• Supplementary file 4. Mtriage summary of the map quality analysis.
DOI: https://doi.org/10.7554/eLife.48496.033

• Transparent reporting form
DOI: https://doi.org/10.7554/eLife.48496.034

### Data availability

CryoEM Density maps and atomic models that support the findings of this study have been deposited in the Electron Microscopy Database and the Protein Databank with the accession codes EMD-4461 (Whole particle reconstruction), EMD-4462 (Vertex Complex), EMD-10237 (Localized reconstruction of the penton region), EMD-10238 (Focused Classification of the penton region) and PDB ID 6Q5U (Atomic model of the asymmetric unit).

The following datasets were generated:

| Author(s) | Year | Dataset title | Dataset URL | Database and Identifier |
|---|---|---|---|---|
| Hemanth KN Reddy, Marta Carroni, Janos Hajdu, Martin Svenda | 2019 | Vertex Complex of Bacteriophage PR772 | https://www.ebi.ac.uk/pdbe/entry/emdb/EMD-4462 | Electron Microscopy Data Bank, EMD-4462 |
| Hemanth KN Reddy, Marta Carroni, Janos Hajdu, Martin Svenda | 2019 | High resolution electron cryo-microscopy structure of the bacteriophage PR772 | https://www.rcsb.org/structure/6Q5U | Protein Data Bank, 6Q5U |
| Hemanth KN Reddy, Marta Carroni, Janos Hajdu, Martin Svenda | 2019 | High resolution electron cryo-microscopy structure of the bacteriophage PR772 | https://www.ebi.ac.uk/pdbe/entry/emdb/EMD-4461 | Electron Microscopy Data Bank, EMD-4461 |
| Hemanth KN Reddy, Marta Carroni, Janos Hajdu, Martin Svenda | 2019 | Focused Classification of the vertex region of Bacteriophage PR772 showing the heteropentameric penton | https://www.ebi.ac.uk/pdbe/entry/emdb/EMD-10238 | Electron Microscopy Data Bank, EMD-10238 |
| Hemanth KN Reddy, Marta Carroni, Janos Hajdu, Martin Svenda | 2019 | Localized Reconstruction of the vertex region of Bacteriophage PR772 showing the heteropentameric penton | https://www.ebi.ac.uk/pdbe/entry/emdb/EMD-10237 | Electron Microscopy Data Bank, EMD-10237 |

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
