## [Decision Letter]

Thank you for submitting your article "CryoEM of coliphage PR772 reveals the composition and structure of the elusive vertex complex and the capsid architecture." for consideration by *eLife*. Your article has been reviewed by three peer reviewers, including Sjors HW Scheres as the Reviewing Editor and Reviewer #1, and the evaluation has been overseen by Olga Boudker as the Senior Editor. The following individuals involved in review of your submission have agreed to reveal their identity: Carmen San Martin (Reviewer #2); Brenda Hogue (Reviewer #3).

The reviewers have discussed the reviews with one another and the Reviewing Editor has drafted this decision to help you prepare a revised submission.

Summary:

In this manuscript, the authors present the cryoEM structure of PR772 – a large, tailless, membrane containing bacteriophage. PR772 belongs to the same family as PRD1 (*Tectiviridae*), whose structure was solved at ~4 Å resolution by crystallography in 2004. Here the resolution is much higher (2.75 Å on average), allowing a more detailed analysis of the interactions in the capsid. The main findings are the heteropentameric organization of the penton capsomer, formed by the small single jelly roll protein P31 and its homolog P5 N-terminal domain (P5N), which is prolonged into a trimeric spike; a low resolution map of the penton organization together with its double spike, formed by the P5C trimer and the monomeric P2 protein; and a detailed description of the alternate conformations of N- and C-termini in the different copies of the major coat protein P3, as well as in the two penton proteins P31 and P5N

The resolution is impressive, particularly for such a large specimen, and addresses important issues in complex virus assembly, such as the use of flexible regions to achieve a range of quasi-equivalent interactions in the capsid, and the long standing puzzle on the composition and organization of the heterooligomeric vertex capsomer and its double spike. Therefore, the subject is interesting and the quality of the work is high. However, the manuscript is poorly written, and the most interesting results of a P31/P5 hetero-pentamer and a monomeric copy of P2 binding to P5 are based on extremely poor reconstructions, which are possibly caused by incorrect processing. Therefore, this manuscript would need considerable modification before being suitable for publication.

Essential revisions:

1) The asymmetric reconstructions of the hetero-pentamer (Figure 4F) and the monomeric P2 are very low-resolution compared to the icosahedral reconstruction. This is probably the result of incorrect image processing. The icosahedrally-averaged pentamer has resolutions extending well beyond 3 Å. Just by lowering the number of asymmetric units by 5-fold, the drop in resolution of the asymmetric reconstructions is too large. By keeping the orientations the same as in the original icosahedrally refined structure (i.e. by doing focussed classifications on the symmetry-expanded particles without re-alignments), resolutions around 3 Å should still be achievable in the asymmetric reconstructions. A good control is to perform a reconstruction of the symmetry expanded (partial signal subtracted) images without any classification/realignment, which should give a map that is almost identical to the icosahedrally averaged reconstruction. The authors should fix this mistake in a revised version, as it would yield a much higher-resolution, and thus more interesting, view on the asymmetry in the hetero-pentamer. In addition, the authors should present detailed views of the improved density, including side-chains, for regions where the sequence between P5 and P31 differ, which would then replace the ambiguous tracing of both proteins in the icosahedrally averaged map at two different thresholds in the current manuscript.

To some extent, the same argument also holds for the P2 monomer reconstruction: doing purely focussed classifications (without re-alignments) of symmetry-expanded images should also improve this part of the map. The only caveat (and cause for lower resolution) in this case would be different orientations of the P2 monomer relative to the capsid. However, even this may be captured better by focussed classifications without alignment than by the currently used focussed refinements.

2) The paper would benefit greatly from extensive editing of the text, while carefully checking for grammar and spelling. In addition, the manuscript organization needs to be improved:

2.1) Previous knowledge hinting at the possible penton heteropentamer/double spike composition and organization is cited, but a clear account of the subject should appear in the Introduction. For instance: "many models have been proposed"… why? The authors should explain that the basis for these previous models is the sequence similarity between proteins P31 and P5 (see for example, Merckel et al., 2005; Huiskonen et al., 2007; Abrescia et al., 2004).

2.2) Also lacking is the information on how similar PRD1 and PR772 are at the genome level (Introduction, second paragraph, repeated in the third paragraph: "a close relative", how close? Should we expect large differences?).

2.3) The two alternative conformations of the P5 N-terminal tail are described twice: first in the third paragraph of the subsection “Penton base is a heteropentamer of P5 and P31”, then in the fourth paragraph of the subsection “Overall Architecture of PR772”.

2.4) The description of the different conformations of the P3 termini in the subsection “Major Capsid Protein and its conformations” is difficult to follow. For example, "the helix… is not embedded deeply into the adjacent subunit of the trimer…": what is meant here? And "the loop formed by Tyr351-Val358 is flipped…": flipped how? Perhaps a superposition of the twelve P3 monomers in the asymmetric units, zooming on the N- and C-termini, would help to better illustrate the mobility of these regions. In general, adding labels to the figures would help the reader to follow the structure description. For example, for some of the residues cited in the text; or for at least one occurrence of each of the P3 N- and C-terminal conformations in Figure 2.

2.5) Including the length of proteins and of their traced part in the Results section would also help the reader (without having to resort to the PDB validation report).

2.6) The start of the Discussion section reads like a repetition of (parts of) the Results section.

2.7) Some sentences seem to be repeated, or at least very similar, e.g. in the paragraph just before Figure 3 and just after Figure 3. Other pieces of text read more like an early draft than like a polished manuscript, e.g. the second paragraph of the subsection “Overall Architecture of PR772”.

---

## [Author Response]

Essential revisions:1) The asymmetric reconstructions of the hetero-pentamer (Figure 4F) and the monomeric P2 are very low-resolution compared to the icosahedral reconstruction. This is probably the result of incorrect image processing. The icosahedrally-averaged pentamer has resolutions extending well beyond 3 Å. Just by lowering the number of asymmetric units by 5-fold, the drop in resolution of the asymmetric reconstructions is too large. By keeping the orientations the same as in the original icosahedrally refined structure (i.e. by doing focussed classifications on the symmetry-expanded particles without re-alignments), resolutions around 3 Å should still be achievable in the asymmetric reconstructions. A good control is to perform a reconstruction of the symmetry expanded (partial signal subtracted) images without any classification/realignment, which should give a map that is almost identical to the icosahedrally averaged reconstruction. The authors should fix this mistake in a revised version, as it would yield a much higher-resolution, and thus more interesting, view on the asymmetry in the hetero-pentamer. In addition, the authors should present detailed views of the improved density, including side-chains, for regions where the sequence between P5 and P31 differ, which would then replace the ambiguous tracing of both proteins in the icosahedrally averaged map at two different thresholds in the current manuscript.To some extent, the same argument also holds for the P2 monomer reconstruction: doing purely focussed classifications (without re-alignments) of symmetry-expanded images should also improve this part of the map. The only caveat (and cause for lower resolution) in this case would be different orientations of the P2 monomer relative to the capsid. However, even this may be captured better by focussed classifications without alignment than by the currently used focussed refinements.

The lower resolution map from the localized asymmetric reconstruction was due to the following reasons: we used 2x binned particles for the localized asymmetric reconstruction. The 2x binned particles have an apix of 2.12 Å, resulting in a Nyquist frequency of 0.2358 (i.e., a resolution of 4.24 Å). So, the reconstructions with these 2x binned particles is limited to a resolution of 4.24 Å. We had also limited the e-step to 10 Å during the classification as recommended by the authors of the localized reconstruction method. We avoided using a mask to remove any bias that could be introduced due to the applied masked. Due to the limitations of storage, we were also limited to using sub-particles from one half map.

We have currently improved the resolution of the penton to 4.41 Å using a soft mask around the penton region and not limiting the e-step during the localized asymmetric reconstruction.

Using the focused classification method as suggested, we were able to reproduce the penton map to a resolution of 4.25 Å, a resolution similar to what was obtained by the localized reconstruction method with 2x binned particles. From the localized reconstruction, we knew that all the classes showed the heteropentameric penton. This suggested that the heteropentameric penton was a common feature. So, we chose a random vertex to perform a C1 focused classification on, without re-alignment. The focused classification of the penton region as well as the localized reconstruction method, show a stalk-like protrusion from 3 of the 5 subunits that form the penton. We have added a new Figure 4—figure supplement 6, showing the maps of the penton region from both methods.

The unmasked localized reconstruction using the sub-particles of the vertex region was performed to determine the orientation of P2 with respect to the viral capsid and avoid any bias by masking. From the localized reconstruction, we now know that the P2 protein is on average present only on 2 vertices of the viral particle. We can also notice that all the vertices display the presence of P5 that would also contribute to map densities (i.e., the highly flexible C-terminal regions with the collagen-like triple helix and the knob domain that are not resolved in the icosahedrally averaged map used to generate the projection for signal subtraction) during the focused classification. These issues along with the caveat mentioned by the reviewers will significantly reduce the resolution. By applying a focus mask around the P2 protein to address some of the issues, we would essentially bias the very orientation that we plan to determine. Hence, we have avoided using whole particles for focused classification of P2. Instead, we provide the P2 map from the masked classification without re-alignment of sub-particles whose orientation was previously determined without using a mask from localized reconstruction. This ensures that the features observed are not due to masking. We have provided two images, one with the map generated by re-aligned sub-particles (previously submitted) and another with a masked map without realignment (submitted here as part of the supporting files). We believe that the Figure 5, currently in the manuscript, illustrates the structure of the vertex complex, best.

We do agree that having a resolution beyond 3 Å for an asymmetric reconstruction would be better to access the heteropentameric nature of the penton. The resolution of the maps from the localized reconstruction and focused classification should be sufficient to assess the secondary structure and domain fit for the proteins P31, P5 N-terminal domain and P2.

2) The paper would benefit greatly from extensive editing of the text, while carefully checking for grammar and spelling. In addition, the manuscript organization needs to be improved:2.1) Previous knowledge hinting at the possible penton heteropentamer/double spike composition and organization is cited, but a clear account of the subject should appear in the Introduction. For instance: "many models have been proposed"… why? The authors should explain that the basis for these previous models is the sequence similarity between proteins P31 and P5 (see for example, Merckel et al., 2005; Huiskonen et al., 2007; Abrescia et al., 2004).

Most of these models were speculated on based on the observations from gene mutation/knock-out and in vitro studies of proteins P31, P5 and P2 from PRD1. The in vitro protein expression studies showed that P31 forms pentamers and the P5 forms multimers of trimers (i.e., (P5_3)1_, (P5_3)2_, (P5_3)3_) (Sokolova et al., 2001). The gene knock-out studies showed that P31^-^ mutants produced incomplete PRD1 particles that lacked P31, P5 and P2 functions. They failed to form the vertex complexes. A P5^-^ mutant produced intact viral particles but lacked both the P5 and P2 functions (Rydman et al., 1999). These observations led to the hypothesis that the vertex complex appears to be a single spike formed by a pentameric P31 base binds the trimeric P5 spike protein to which P2 is bound (Huiskonen et al., 2007; Caldentey et al., 1999; Sokolova et al., 2001). Later, a combination of SAXS modelling of the P5 protein and a low resolution cryoem study of the vertex region, showed that P2 and P5 form two spikes, not one, as previously described. Based on the low-resolution SAXS model of P5, it was speculated that the N-terminal base of P5 could have a similar size and fold to P31 due to sequence similarity (Huiskonen et al., 2007).

We have added the above explanation to the manuscript to clarify why many models were proposed to explain the architecture of the vertex complex.

2.2) Also lacking is the information on how similar PRD1 and PR772 are at the genome level (Introduction, second paragraph, repeated in the third paragraph: "a close relative", how close? Should we expect large differences?).

We have now included the DNA sequence identity between PRD1 and PR772. We have also included the reference (Saren et al., 2005), which describes the genome level similarity between different phages in the *Tectiviridae* family.

“Much of the functional knowledge of the viral proteins is inferred from previous studies on PRD1, a close relative of PR772 with a genome sequence identity of 97.2% (Lute et al., 2004; Saren et al., 2005).”

2.3) The two alternative conformations of the P5 N-terminal tail are described twice: first in the third paragraph of the subsection “Penton base is a heteropentamer of P5 and P31”, then in the fourth paragraph of the subsection “Overall Architecture of PR772”.

In the subsection “Penton base is a heteropentamer of P5 and P31”, we are establishing that it is the P5 N-terminal that is responsible for the special conformation. in the subsection “Overall Architecture of PR772”, we are comparing the N-terminal regions of P5 with P31 and their influence on the overall architecture. We only mention the special conformation of the N-terminal P5 region in one sentence to remind the reader that it exists. This is relevant for the hypothesis we put forward in the discussion regarding the potential infection process.

2.4) The description of the different conformations of the P3 termini in the subsection “Major Capsid Protein and its conformations” is difficult to follow. For example, "the helix… is not embedded deeply into the adjacent subunit of the trimer…": what is meant here? And "the loop formed by Tyr351-Val358 is flipped…": flipped how? Perhaps a superposition of the twelve P3 monomers in the asymmetric units, zooming on the N- and C-termini, would help to better illustrate the mobility of these regions. In general, adding labels to the figures would help the reader to follow the structure description. For example, for some of the residues cited in the text; or for at least one occurrence of each of the P3 N- and C-terminal conformations in Figure 2.

We agree that only text makes it a bit hard to follow the reasoning. As suggested, we have added new figures and have tried to refer to figures everywhere possible to make it easier to understand what is being described in the text. We have added Figure 2—figure supplement 3, that shows the superposition of all the P3 monomers of the asymmetric unit. This figure also includes an image showing the flipped Tyr351-Val358 loop.

2.5) Including the length of proteins and of their traced part in the Results section would also help the reader (without having to resort to the PDB validation report)

We have added the details about the proteins, chain length, length of the chain traced and region, where we compare the current model of PR772 with model of PRD1.

2.6) The start of the Discussion section reads like a repetition of (parts of) the Results section.

We have rephrased the start of the Discussion to only include differences compared to the PRD1 structure.

2.7) Some sentences seem to be repeated, or at least very similar, e.g. in the paragraph just before Figure 3 and just after Figure 3. Other pieces of text read more like an early draft than like a polished manuscript, e.g. the second paragraph of the subsection “Overall Architecture of PR772”.

In these two instances (in the paragraphs before and after Figure 3), we are describing different things.

In the paragraph before Figure 3, we describe that the N-terminal of P3 can adopt three different conformations not two as previously described for PRD1. The new conformation as compared to PRD1 can only be observed due to the higher resolution model/reconstruction and has relevance for the detailed molecular interplay between proteins building up the capsid.

In the paragraph after Figure 3 and onwards, we describe how the N and C-terminals of the “same” P3 subunit that form four different trimers have to adopt different conformations depending on where the trimer resides in the trisymmetron and what contacts with other proteins they make at that specific location.

We have polished the text. It now reads:

“Five copies of P16 bridge the penton to the five trisymmetrons that meet at the vertex complex. […] As was mentioned earlier, to accommodate the P30 protein in the P3-P30-P16 complex (Figure 6 and Figure 6—video 1), the loop formed by residues Tyr351-Val358 of the P3 subunit in trimer 1 is flipped compared to the orientation seen in this loop of the P3 subunit with a N-terminal long helix (Figure 2—figure supplement 3D-F).”